# BUILD-BENCH: BENCHMARKING LLM AGENTS ON COMPILING REAL-WORLD OPEN-SOURCE SOFTWARE

## ABSTRACT

Automatically compiling open-source software (OSS) projects is a vital, labor-intensive, and complex task, which makes it a good challenge for LLM Agents. Existing methods rely on manually curated rules and workflows, which cannot adapt to OSS that requires customized configuration or environment setup. Recent attempts using Large Language Models (LLMs) used selective evaluation on a subset of highly rated OSS, a practice that underestimates the realistic challenges of OSS compilation. In practice, compilation instructions are often absent, dependencies are undocumented, and successful builds may even require patching source files or modifying build scripts. We propose a more challenging and realistic benchmark, BUILD-BENCH, comprising OSS that are more diverse in quality, scale, and characteristics. Furthermore, we propose a strong baseline LLM-based agent, OSS-BUILD-AGENT, an effective system with enhanced build instruction retrieval module that achieves state-of-the-art performance on BUILD-BENCH and is adaptable to heterogeneous OSS characteristics. We also provide detailed analysis regarding different compilation method design choices and their influence to the whole task, offering insights to guide future advances. We believe performance on BUILD-BENCH can faithfully reflect an agent's ability to tackle compilation as a complex software engineering tasks, and, as such, our benchmark will spur innovation with a significant impact on downstream applications in the fields of software development and software security.

## 1 INTRODUCTION

Imagine that you are a graduate student during a rebuttal period. The reviewers strongly suggested that you compare your system with prior work. It was only published a few years ago, so you find the GitHub repo, download the code, and read the included docs. It doesn't compile. The dependency URLs are missing. Required libraries aren't included. Even if it worked perfectly when first published, it's going to take you days to even compile this system. This scenario highlights the difficulty of compiling once-maintained open-source code, and in fact this problem is faced by the broader software engineering community. However, recent advances in Large Language Models (LLMs) promise to improve various software engineering tasks (Brown et al., 2020; Touvron et al., 2023; Chen et al., 2021). While commercial software can be developed with stringent and consistent engineering practices, OSS projects are highly heterogeneous. Additionally, OSS projects are maintained by varied contributors, adopt various build frameworks, and frequently require platform-specific configurations.

Compiling OSS often requires manual intervention to resolve missing dependencies, version mismatches, or undocumented environment requirements. Although prior rule-based methods (e.g., GHCC (Hu, 2020) and Assemblage (Liu et al., 2024)) attempted to automate this process by iteratively invoking build scripts, they cannot robustly handle dependency, toolchain, or platform mismatches. These challenges impact human software engineers who integrate OSS into their own applications, as this requires compilation to turn the software into a library or binary that they can use in their own system. Reliable and scalable automated compilation, in addition to improving software engineering, has other research benefits: It enables large-scale usage of binary data sources, supports program analysis and vulnerability discovery, and accelerates software maintenance workflows (Lacomis et al., 2019; Dramko et al., 2023; Pal et al., 2024). This work addresses these challenges by using LLM-based agents to automate OSS compilation at scale.

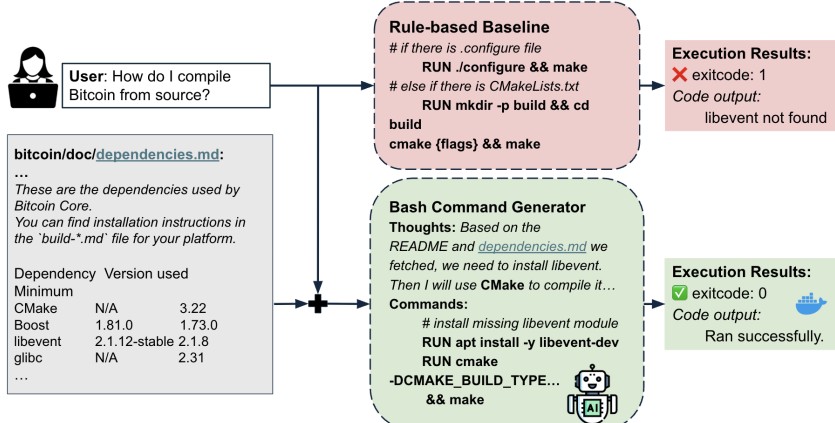

Figure 1: Demonstration of rule-based and AI agentic compilation methods. While rule-based methods follow a predefined workflow, they cannot adequately adapt to different environments. In comparison, AI agents leverage their pre-trained knowledge to adjust the compilation commands based on execution results. In this example, the agent realizes LIBEVENT is a key missing dependency for Bitcoin to compile and installs it to successfully compile the project.

LLMs that are pre-trained on a large amount of natural language data exhibit strong performance in general-purpose tasks, even with zero-shot prompting (Kojima et al., 2023). This capability is generalized to software engineering-related tasks, such as code generation, software debugging, documentation generation, and code refactoring. As such, LLM-based AI agents (Yao et al., 2023; Shinn et al., 2023), which are autonomous systems that use LLMs to iteratively plan, reason, and act, are increasingly used to automate and facilitate complex software engineering workflows (Wang et al., 2024). In this context, we position OSS compilation as a challenging and underexplored target for LLM-based agents. We are thus motivated to create a benchmark (BUILD-BENCH) and systematically evaluate various, specifically agentic, solutions. BUILD-BENCH includes 148 humanly verified repositories out of 385 randomly selected C/C++-heavy OSS from GitHub, each manually annotated for compilation success and build instruction retrieval. We use an additional 70 carefully chosen projects as a validation set to support the development of our agentic baseline method, OSS-BUILD-AGENT. Using BUILD-BENCH, we evaluate existing rule-based and LLM baselines and agentic compilation methods. We showcase the current shortcoming of rule-based methods in compilation performance and success verification. For LLM and agentic compilation methods, we inspect in detail the performance discrepancy of various compilation system designs. Specifically, we demonstrate the effectiveness of our LLM-Assisted Retrieval and Multi-Agent Compilation module design through a side-by-side comparison to another agentic solution. Through the release of BUILD-BENCH and our analysis, we encourage researchers to create better agentic solutions for the compilation task, which will ultimately benefit the AI, software engineering, and software security communities.

***Contributions.*** Our contributions are as follows:

- We created BUILD-BENCH, which is a benchmark that contains both a hand-picked validation set and a randomly sampled test set with manual inspection and labeling to support a rigorous and systematic evaluation of different OSS compilation techniques.

- We evaluated the performance of five rule-based and AI build methods on BUILD-BENCH, including two agentic methods. Our proposed OSS-BUILD-AGENT achieved the best performance, surpassing the strongest rule-based baseline by roughly 50%. With a strong base LLM, OSS-BUILD-AGENT reached a 66.4% success rate, establishing a strong baseline performance, while BUILD-BENCH remains a challenge for future research.

- We offered a detailed inspection of various design approaches in compilation instruction retrieval and error resolution modules and their effects on task performance.

## 2 CONSTRUCTING BUILD-BENCH

A benchmark for automated OSS compilation requires a representative dataset of OSS. We first analyze the prior work COMPILEAGENTBENCH (Hu et al., 2025), which also targets the automatic compilation task. Specifically, it consists of 100 popular and well-known GitHub projects, averaging over 8,000 stars. However, this focus on popular repositories overlooks the vast majority of OSS: 99.88% of C/C++ projects on GitHub have fewer than 500 stars. Therefore, the generalizability of evaluation results on COMPILEAGENTBENCH may be undermined by projects that are unusually well documented, well maintained, and less representative of the in-the-wild challenges.

**Data Filtering.** Therefore, we strive to create BUILD-BENCH as a statistically representative benchmark to ensure generalizability to the broad diversity of OSS. To create the raw dataset, we collected 2.77M C and 4.23M C++ repositories from GitHub RESTAPI with a date range between April 1st, 2008 to January 1st, 2024. To remove extremely low-quality repositories, we apply a few filters: We exclude projects with names or descriptions that contained certain keywords (e.g., `homework` or `assignment`, more in Appendix A) or have a stargazer count below 50 to ensure the OSS are meaningful for both practical usage and research purposes. For deduplication, we excluded repositories that are forks of other repositories. After filtering, the raw dataset contains 6.57M repositories. From this population, we randomly select 385 projects, the minimum sample size required to measure a population proportion with 95% confidence with a margin of error of 5%, according to Cochran (1977) (details in Appendix B). We believe that this randomly sampling helps ensure that BUILD-BENCH better approximates real-world OSS compilation challenges.

**Data Selection and Labeling by Human Experts.** Due to the random sampling process, we cannot guarantee that all of the 385 projects can be compiled. Therefore, human experts manually built each repository to determine its validity. We also exclude OSS repositories that fit into following criteria: (a) The repository targets another operating system and cannot be cross-compiled; (b) The repository only contains trivial or unbuildable content; (c) The repository is missing critical source files and broken dependencies that cannot be installed or created; (d) There are compilation and linking errors that human experts cannot resolve in a best-effort setting.

This process resulted in 148 compilable repositories' names as the final test set. More details can be found in Appendix Section J. During experiments, each repository is freshly cloned at runtime to ensure the code remains unmodified and consistent with its original source. Additionally, we manually created ground truth labels for compiled binary file names and URLs where the build instructions are hosted, if provided by the developers. The manual labeling involved 12 graduate students with more than 3 years of experience working on system research and took roughly 150 hours.

We also created a validation set of 70 popular repositories used to evaluate OSS-BUILD-AGENT.

The representativeness (or diversity) of a benchmark for compilation task is essential to evaluate the generalizability and performance of any compilation technique. We analyze the representativeness from two following aspects: popularity distribution and build system distribution.

**Popularity.** The number of Stargazers (or stars) of a GitHub repository is often used to approximate popularity and perceived quality (Dramko et al., 2023). A higher number often correlates with popular or essential functionality, better code quality, and an active development community that supports continuous and frequent maintenance. However, a majority of repositories tend to have relatively lower stars compared to high-profile projects such as OpenSSL or FFmpeg. This is partially because repositories are often created for specific use cases and targeting smaller and specialized audience groups instead of having widely applicable use cases. Meanwhile, most repositories are for personal or experimental use, further undermining their limited visibility.

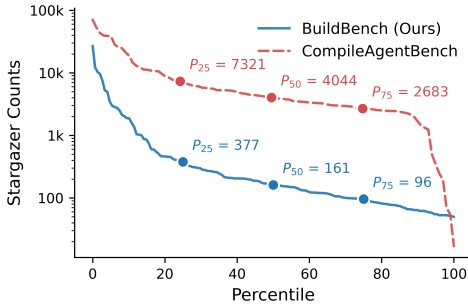

Figure 2: Distribution of stargazer counts of BUILD-BENCH and COMPILEAGENTBENCH. In BUILD-BENCH, relatively low-profile repositories made up majority of the dataset.

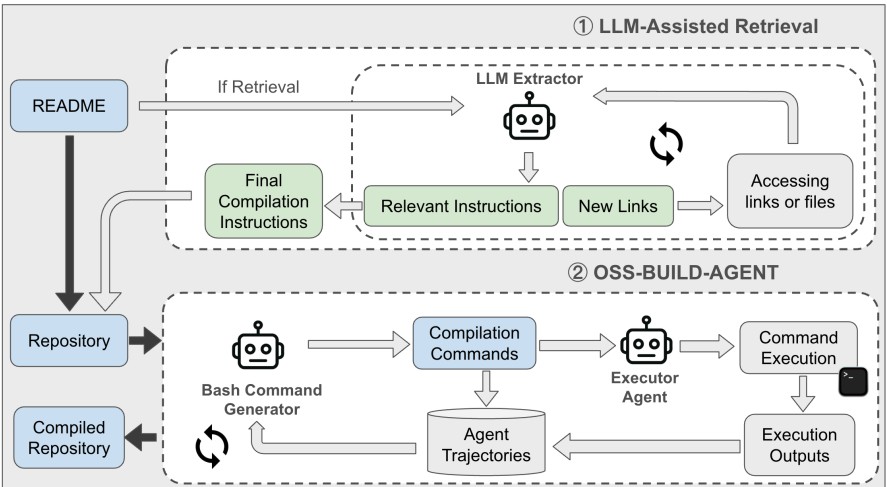

Figure 3: OSS-BUILD-AGENT system diagram. The initial input is the README, then an optional LLM extends this with additional compilation instructions. Finally, a multi-agent build system iteratively generates and executes compilation steps, attempting to compile the target repository.

Figure 2 shows the Stargazer counts of repositories in BUILD-BENCH and COMPILEAGENTBENCH.
Most repositories in BUILD-BENCH are in the 50–500 range, indicating random selection results coincide with the long-tail distribution of overall repository popularity. This characteristic makes BUILD-BENCH more challenging for evaluating compilation techniques, because low-profile repositories often lack documentation or require additional customization or configuration. In contrast, the Stargazer counts of COMPILEAGENTBENCH repositories are aggregated between 2k and 10k, and these popular repositories might be considered as an underestimation of the true difficulty of the compilation task.

**Build Systems.** We further analyze the build systems and tool chains used in BUILD-BENCH repositories: 62 projects use `Make`, 60 use `CMake`, 29 use `Autotools`, and 14 use Visual Studio (`MSBuild`). Smaller—but non-negligible subsets—adopt alternative systems such as custom scripts, `QMake`, `Meson`, etc. 10 repositories provide no explicit build system, often relying on direct compilation. This diversity showcases the heterogeneity of real-world OSS, where the build system selection often depends on the project domain, platform, and community preference. Note that there may be multiple build systems available in the same OSS, which offers alternative compilation approaches.

Overall, the results show that BUILD-BENCH adequately represents a wide variety of real-world C and C++ projects and is suitable as a benchmark for evaluating of automated build techniques.

## 3  AGENTIC BUILDING METHODS

We create an agentic compilation technique, OSS-BUILD-AGENT. As Figure 3 shows, an initial (and optional) LLM iteratively extends the README with additional compilation instructions, then a multi-agent build system iteratively generates and executes compilation steps.

### 3.1  COMPILATION INSTRUCTION RETRIEVAL

Build scripts such as Makefiles are often lengthy, noisy, and machine-oriented, making them undesirable retrieval targets for both humans and AI agents when seeking clear, actionable build instructions. Many repositories with complex build processes or specialized configurations document these steps explicitly for human developers, and retrieving such information provides an effective foundation for agents to generate accurate compilation commands.

However, we find that such instructions is not only located in the OSS repository's README but also in other files inside of repository or on another website. To solve this challenge we propose an LLM-Assisted Retrieval module, an optional component that precedes OSS-BUILD-AGENT.

Our approach uses an out-of-box LLM as an incremental retriever to synthesize the complete set of instructions required for compiling a given repository.

The process uses the project's README as input. The LLM then iteratively performs three operations: (i) it distills potential compilation instructions from the file, (ii) it evaluates the sufficiency of the acquired information, and (iii) if the information is not sufficient enough to support compilation, it identifies promising hyperlinks, encompassing both internal files (of which paths are also parsed as valid URLs to ensure consistency) and external web pages. The contents of up to three newly identified links are subsequently fetched, summarized, and re-evaluated. This recursive process of retrieval and refinement continues until the LLM's confidence in the completeness of the build knowledge is fulfilled or a maximum of three iterations is reached. The output of the retrieval module is the final compilation instructions, which is then passed to the agentic compilation framework.

## 3.2 MULTI-AGENT COMPILATION SYSTEM

The compilation system comprises two cooperating agents, both using an out-of-the-box LLM of the user's selection: *Bash Command Generator* is given the final compilation instructions from the prior module (if using LLM-Assisted Retrieval) and the repository as input and produces a candidate sequence of bash commands to compile the repository. *Execution Agent* runs these commands within a containerized environment and returns the execution results. Prompts are included in Appendix Section C.3 and C.4. For refinement steps $k = 0, \ldots, K$, let $C_k$ be the input into *Bash Command Generator*, which produces $S_k$, the commands to be executed by *Execution Agent* in the environment that returns execution results $f_k$.

**Initialization.** *Bash Command Generator* produces the first set of commands directly from the input prompt $C_0$ because there is no execution feedback. *Execution Agent* runs generated commands $S_0$ in a fresh Docker container, yielding the initial execution results $f_0$.

**Iterative Error Resolution.** This constitutes the standard agentic loop: *Bash Command Generator* uses both $C_k$ and the latest execution results $f_k$ to craft revised commands $S_k$, which the *Execution Agent* then executes again. Note that all potential error information in $f_k$ is resulting from the execution errors of incorrect bash commands generated by agents.

The process ends when $\text{Success}(f_k) = \text{true}$ or when $k = K$, exceeding the maximum turns allowed. This iterative error resolution process enables the compilation to recover from missing dependencies, incorrect flags, or environmental mismatches, an ability that is required for OSS compilation task, as we analyzed in Section 6.

## 4 BASELINE METHODS

In this section, we present existing rule-based techniques as well as two LLM-based compilation methods we compare against.

**GHCC.** GHCC (Hu, 2020) is a rule-based tool for building GitHub repositories. Prior research uses datasets that GHCC created (Lacomis et al., 2019; Xie et al., 2024). Given a repository, GHCC attempts to build the project by first discovering all build system-specific files (e.g., Makefile and CMakeLists.txt) and then conducting a rule-based build routine customized for these build systems.

**Assemblage.** Assemblage (Liu et al., 2024) is a system designed to automate the construction of binary datasets of primarily Windows executables by building source code. It follows a similar rule-based compilation workflow as GHCC.

**Single-turn LLM baseline.** To evaluate the project-building performance of pretrained LLMs on a single-turn basis, we prompt an out-of-the-box LLM to generate a set of Bash commands to build a target repository and execute the commands in a Docker container. The input to this baseline is the README file and file directory of the OSS's root directory. Without any execution feedback, this single-turn baseline cannot adjust its initial output. (Prompt in Appendix C.2.)

**CompileAgent.** CompileAgent (Hu et al., 2025) also introduces a multi-agentic compilation system. It adopts a flow-based agent strategy in which a master agent orchestrates the build process across two core modules: (1) CompileNavigator for locating and extracting build instructions and (2) ErrorSolver for resolving compilation errors. These modules are supported by five specialized tools (shell execution, file navigation, instruction extraction, web search, and multi-agent discussion), four of which involve auxiliary LLM agents, totaling seven agents in the pipeline. We include its official open-source implementation as a baseline in our evaluation to provide a representative comparison against our agentic approaches.

## 5 EXPERIMENT SETUP AND EVALUATION METHODS

We evaluate the performance of baseline build techniques and OSS-BUILD-AGENT on BUILD-BENCH. We implement single-turn LLM baseline with two base models: GPT o3-mini and Claude 3.7-Sonnet. For OSS-BUILD-AGENT, we use five models, representing diverse characteristics including reasoning vs. non-reasoning, generic vs. coding-specific, and different parameter sizes. For CompileAgent we use GPT-4o as the main base model as its implementation indicates. All build methods build each repository in a fresh Ubuntu 22.04 Docker container, with minimal packages pre-installed.

**Success Metrics.** A key evaluation challenge is to determine if a build method successfully builds a given repository. Existing build methods determine the compilation process as **Completion** with the presence of at least one binary post-building. This metric is unreliable when (1) a failed building process generates intermediate binary files, or (2) a submodule (or a vendored package) successfully builds while the main repository fails building.

We improve the completion success criteria with additional validation using expert-generated, per-repository lists of binary file names as ground truth. After the building of a repository completes, we compare the file names of all produced binary files against a expert-generated list. We categorize success into two types: (1) **Strict Success** only when all binary file names in the expert-generated list exist, and (2) **Flexible Success** when at least one file name in our expert-generated list exists.

## 6 EVALUATING BUILD METHODS

Table 1 presents the performance of all build methods on BUILD-BENCH.

**Baselines.** For rule-based methods, GHCC achieves 30.2% completions and 13.4% flexible validated successes, outperforming Assemblage. Single-turn LLM baselines' results vary: o3-mini exhibits degraded performance, while Claude 3.7-Sonnet is surprisingly strong for a non-agent setting (21.5% strict; 22.1% flexible). Moreover, the performance of COMPILEAGENT suffers a substantial drop from 89% strict validated success on COMPILEAGENTBENCH to 49.7% strict and 55.7% flexible on BUILD-BENCH. This performance drop indicates a pronounced distribution shift and higher difficulty of BUILD-BENCH.

**Agents enable compilation error resolution in multi-turn setting.** OSS-BUILD-AGENT substantially outperforms all rule-based baselines. The best configuration, OSS-BUILD-AGENT with LLM-assisted Retrieval using Claude 3.7-Sonnet, reaches 66.4% strict and 71.8% flexible validated successes, a gain of 49.7 percentage points over single-turn baseline with the same model. Iterative observation–repair–rebuild loops allow agents to access and receive feedback from execution results, backtrack from ineffective commands, and apply targeted fixes that single-turn approaches cannot.

**Agentic build methods are model-agnostic, but scale with model intelligence.** The agent framework uses out-of-the-box pre-trained LLMs, allowing our framework to be model-agnostic. Nevertheless, performance scales with model capability. Among all settings, Claude 3.7-Sonnet achieves the best performance with significant margin to the next best model. This confirms that stronger LLMs are more effective in adjusting its output based on error results and applying targeted fixes, two skills that are central to resolving complex build failures. In contrast, smaller models (e.g., o3-mini) perform consistently but saturate at around 68–69% flexible success, while specialized models (Qwen3 Coder) underperform (38.9% flexible), suggesting that coding specialization may become a drawback, considering retrieval module challenges more on model's documentation comprehension

Table 1: Performance of all evaluated build techniques on BUILD-BENCH test set. Section 5 describes the evaluation metrics of completion and validated successes.

| LLM Usage | Build Method | Un-validated Completions % | Validated Successes % | |
|---|---|---|---|---|
| | | | *Strict* | *Flexible* |
| N/A | GHCC | 30.2 | 10.1 | 13.4 |
| N/A | Assemblage | 10.7 | 6.0 | 9.4 |
| Single Turn | *LLM Baseline* | | | |
| | o3-mini | 63.5 | 33.1 | 36.5 |
| | GPT-4o | 60.8 | 30.4 | 32.4 |
| | Claude 3.7-Sonnet | 64.9 | 37.2 | 39.9 |
| | Gemini 2.5 flash | 60.8 | 34.5 | 37.2 |
| | Qwen3 235B | 62.2 | 33.8 | 35.8 |
| | Qwen3 Coder 480B | 65.5 | 35.8 | 38.5 |
| Multi-Agents | CompileAgent (GPT-4o with Retrieval) | N/A | 50.7 | 56.8 |
| Multi-Agents | **OSS-BUILD-AGENT *w/o Retrieval (Ours)*** | | | |
| | GPT-4o | 56.8 | 38.5 | 41.9 |
| | o3-mini | 67.6 | 48.0 | 50.7 |
| | Claude 3.7-Sonnet | 83.1 | 64.1 | 70.9 |
| | Gemini-2.5-flash | 75.0 | 55.4 | 60.1 |
| | Qwen3 235B | 80.4 | 58.8 | 63.5 |
| | Qwen3 Coder 480B | 84.5 | 57.4 | 61.5 |
| Multi-Agents | **OSS-BUILD-AGENT *w/ LLM-Assisted Retrieval (Ours)*** | | | |
| | GPT-4o (Avg of 3 Runs) | 70.2 | 53.0 | 57.6 |
| | o3-mini | 79.9 | 63.1 | 68.5 |
| | **Claude 3.7-Sonnet** | **85.2** | **67.6** | **73.0** |
| | Gemini-2.5-flash | 77.2 | 58.1 | 62.2 |
| | Qwen3 235B | 83.9 | 60.8 | 67.6 |
| | Qwen3 Coder 480B | 48.3 | 34.2 | 38.9 |

ability. Overall, the performance of OSS-BUILD-AGENT is model-agnostic, but stronger LLMs still improve the performance of OSS-BUILD-AGENT.

## 6.1 INSTABILITY AND REPEATED EXPERIMENTS

Table 2: Results from three repeated runs of OSS-BUILD-AGENT with retrieval using GPT-4o. $k$ refers to the order in Figure 4. Error Fixing attempts is the average of attempts across all repositories in one run.

| k | Error Fixing Attempts | Strict Success % | Flexible Success % |
|---|---|---|---|
| $k = 2$ | 4.8 | 45.6 | 50.3 |
| $k = 1$ | 6.9 | 54.7 | 59.5 |
| $k = 3$ | 8.4 | 58.8 | 62.9 |

Instability in agentic frameworks is a well-recognized issue (Yao et al., 2024). Although OSS-BUILD-AGENT performs strongly, its results fluctuate over runs. To quantify this, we repeat experiments with GPT-4o, a non-reasoning model, as the base model in three independent runs. Table 2 shows the results, where OSS-BUILD-AGENT achieves 53.0% ± 6.8 strict and 57.6% ± 6.5 flexible validated success, indicating non-trivial variance. We attribute this to two major factors. First, LLM-guided retrieval can follow different documentation accessing trajectories and produce different build recipes across runs, shifting the subsequent compilation trajectories. Second, LLM outputs are non-deterministic even with identical prompts (Song et al., 2024), and this randomness compounds over multi-turn interactions. Together, these effects lead to instability.

Additionally, we evaluate pass@k across three runs to assess the benefit of multiple attempts (Figure 4). For the strict setting, the pass rates increase from 54.7% at pass@1 to 59.5% at pass@2 and 65.5% at pass@3. Under the flexible setting, the corresponding rates are higher, rising from 59.5% to 64.2% and 70.3%. These results demonstrate that multiple agentic trials substantially improve performance, which may better control the stochastic nature of AI agents. Repeated experiments not only control for performance variance, but also help to validate the arguments based on performance.

## 6.2 RETRIEVAL AND ERROR RESOLUTION

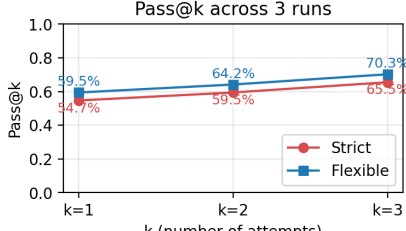

Despite the architectural differences between COM-PILEAGENT and OSS-BUILD-AGENT, they both incorporate two similar modules of build instruction retrieval and agentic error resolution. We discuss the system design differences and their performance impact.

**Retrieval Performance Analysis.** Accurate retrieval of build instructions has a strong impact on subsequent compilation performance. Developer-provided instructions offer a solid starting point that agents can adapt to match specific configuration requirements or environment differences. In BUILD-BENCH, we identified 136 OSS repositories from BUILD-BENCH test set with clear URL labels

Figure 4: pass@k performance of OSS-BUILD-AGENT with LLM-Assisted Retrieval using GPT-4o. For $K = 1$, we report the earliest chronological run.

for the build instruction. Together, these form a secondary benchmark for evaluating the retrieval module described in Section 3.1.

We evaluated COMPILEAGENT and OSS-BUILD-AGENT's LLM-Assisted Retrieval (both using GPT-4o) on the secondary retrieval benchmark. The criterion for success is whether the retrieval module accesses the ground-truth URL, which may correspond to either an internal file within the repository (e.g., parsed as a GitHub URL) or an external website that hosts the build instruction for the given repository. We conducted additional experiments with respect to the sensitivity of base model choices (Table 4) and different retrieval strategies such as TF-IDF and RAG (Table 5). Further analysis can be found in Appendix Section D.

In our evaluation, the retrieval module of OSS-BUILD-AGENT achieved a retrieval accuracy of 73.8%, significantly outperforming COMPILEAGENT's 46.2%. We attribute this performance improvement to key design choices in our retrieval module.

We observe that COMPILEAGENT's retrieval tool favors certain files or pages and often avoids less obvious links, leading to missed instructions. For example, when given the structure of the root directory of a repository, agents usually pick build scripts (e.g., `Makefile`) as the retrieval target. Unfortunately, build scripts are often too noisy and can divert the agents from continuing to find explicit documentation about configuration or setup. Additionally, build instructions can exist across multiple sources (e.g., README files, wiki pages, and subdirectories), and the derailment of agents compounds when facing noises from the scattered instructions.

In comparison, we design the LLM-Assisted Retrieval module of OSS-BUILD-AGENT as a workflow that mimics a human engineer. It focuses on exploring the documentation instead of the build process. In the first iteration of retrieval, we instruct the LLM to inspect the main README file to extract information or find useful URLs. This prevents the LLM from being distracted by build scripts. Traversing a path of documentation files, our retrieval module better handles scattered information.

**Retrieval Strategies for Compilation Task.** We evaluate how retrieval strategies affect OSS-BUILD-AGENT's compilation performance (Table 3).

An Retrieval-Augmented Generation (RAG) Lewis et al. (2021) baseline is added for better comparison. The RAG corpus is built upon all internal documentation files and a README-seeded three-hop link crawl. We first extract all URLs referenced in the repository's README (depth 1). For each fetched page, we extract its outgoing URLs and repeat this expansion twice more (depths 2 and 3). We index the content of all pages at depths 1–3. We also report a *Perfect Retrieval* ablation that adds the ground-truth build instructions directly to the OSS-BUILD-AGENT' prompt.

All retrieval variants improve over the *No Retrieval* baseline. LLM-Assisted Retrieval consistently outperforms RAG in all metrics by a significant margin, with 4.3% more in the strict validated success percentage. It shows that targeted LLM-guided selection can provide agents with build signals of higher-quality than RAG. As expected, Perfect Retrieval sets the upper bound on validated correctness. It suggests that accurate retrieval of that human-written build instructions can drastically improve performance rather than merely increasing retrieval coverage. In general, these results confirm that the integration of retrieval is necessary for robust compilation and that the precision of the retrieval is the key factor of the validated gains.

Table 3: Ablation of retrieval strategies for OSS-BUILD-AGENT. Values are percentages; parentheses show relative change vs No Retrieval baseline. Base model is o3-mini.

| Retrieval Method | Unvalidated (%) | Validated Success (%) | |
| --- | --- | --- | --- |
| | | *Strict* | *Flexible* |
| OSS-BUILD-AGENT – **No Retrieval** (Baseline) | 67.6 | 48.0 | 50.7 |
| OSS-BUILD-AGENT – **LLM-Assisted Retrieval** | 79.9 (+12.3) | 63.1 (+15.1) | 68.5 (+17.8) |
| OSS-BUILD-AGENT – **RAG** (Ablation) | 76.4 (+8.8) | 58.8 (+10.8) | 64.9 (+14.2) |
| OSS-BUILD-AGENT – **Perfect Retrieval** (Ablation) | 79.1 (+11.5) | 68.2 (+20.2) | 70.9 (+20.2) |

**Error Resolution Attempts.** We compare two different agentic systems and manually inspect their action trajectories of error resolution. We observe that while COMPILEAGENT employs a variety of tools, the main agent rarely invokes some of these tools (such as Multi-Agent Discussion for error resolution). The master agent usually exits too "easily" when encountering compilation errors, without attempting more fixes by invoking tools. Because compilation errors are often long, verbose, and nested, locating and fixing root-cause errors may require iterative attempts (interested readers may refer to an example in Appendix H). Thus, more error resolving attempts is favorable, which is validated by our repeated experiments with OSS-BUILD-AGENT as Table 2 shows. Using the same base model, we observe that validated success rate scales well with the number of attempts to resolve the error.

Despite the scaling effect of error resolution attempts, OSS-BUILD-AGENT attempts 6.6 times on average, in comparison to 7.5 times in COMPILEAGENT (excluding its retrieval module for fair comparison). This difference is due to our agent outputting the entire set of build commands, while COMPILEAGENT outputs one bash command at a time and refines it iteratively if execution shows error. While this fine-grained approach can be effective, it also inflates the trajectory with trivial commands (e.g., ls, mkdir) that rarely fail but still count as separate steps. Conversely, OSS-BUILD-AGENT generates a more complete set of compilation commands intended to drive the build to completion in a single run, followed by troubleshooting if needed. This design allows the agent to observe the full command history at each step, providing contextual information for error resolution. For example (details in Appendix G), an error such as *The source directory does not appear to contain CMakeLists.txt* can be resolved more effectively when the agent has access to prior directory navigation steps, enabling it to adjust the working directory and retry seamlessly.

Together, the agentic design in OSS-BUILD-AGENT enables effective retrieval and error resolution, achieving superior performance with two agents (compared to seven in COMPILEAGENT) and a simpler architecture. This demonstrates the competitive efficiency of our end-to-end agentic compilation pipeline.

### 6.3 FAILURE MODES OF AGENTIC BUILD METHODS

Agentic methods are known for instability, task derailment, disobeying instructions, and many other drawbacks (Cemri et al., 2025). Thus, it is important to identify the failure modes of the agents to facilitate the future development of more potent agents for compilation task. We manually inspect the building process executed by OSS-BUILD-AGENT, with base model being o3-mini and enhanced with the LLM-assisted retrieval approach. The results are shown in Figure 5.

The most common failure arises from missing packages or version mismatches. Modern build systems typically validate dependency integrity before compilation, so unresolved package constraints would break compilation task early. Yet we observe that agents do not possess enough information regarding specific versions or constrains of specific dependencies, preventing them to resolve the issue.

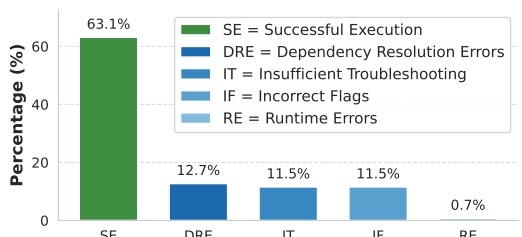

Figure 5: Agent failure modes analysis of OSS-BUILD-AGENT enhanced with LLM-Assisted Retrieval implemented with o3-mini.

Insufficient troubleshooting is another noticeable failure mode for agentic compilation methods. Complex repositories often require several rounds of incremental repairs (e.g., addressing transitive headers, linker flags, or generator invocations), but agents may quit early when determining the build is impossible after a few attempts. Moreover, another pattern is the under-exploration of build alternatives. Some projects support multiple build systems (e.g. cmake and a customized setup script), but agents may not explore all the possible methods before failing on one method and quitting the work.

Occasionally the agent proposes a plausible bash command, but in which the flags are ordered incorrectly or partially fabricated. Because link order and library grouping are order-sensitive in common toolchains, such mistakes will produce compilation or linking errors even when the base command is correct.

## 7 RELATED WORK

LLM has shown promising performance across various software engineering tasks. These include automated resolution of GitHub issues (Jimenez et al., 2024; Su et al., 2025), intelligent code generation (Ishibashi and Nishimura, 2024), automated test case generation (Pizzorno and Berger, 2025; Yuan et al., 2024), and Python software installation (Milliken et al., 2024). Within this growing landscape, the task of automatically compiling C/C++ OSS remains relatively underexplored. The intricacies of these languages, including discontinued maintenance, complex build systems that depend on many external dependencies, and the often less informative error messages from compilers like GCC and Clang (Onyango and Mariga, 2023), all add up to the difficulties of the task. Rule-based methods have been used extensively in previous work on building binary datasets for downstream tasks (Hu, 2020; Lacomis et al., 2019; Liu et al., 2024). While such methods suffer from their inherent fragility, AI agents may be a suitable solution. As initial efforts, such as CompileAgent (Hu et al., 2025), have indicated the potential of using the agentic compilation method, the dataset against which it is evaluated consists of many well-known OSS that may have their compilation processes memorized by LLM, introducing biases to the evaluation results. We believe it is necessary to create a more challenging and representative benchmark dataset that allows for more insightful evaluation and analysis of agentic compilation methods.

## 8 LIMITATIONS

We evaluate solely on compiling C/C++ repositories on the Linux system, which is beneficial for many downstream tasks (Brust et al., 2023; Arasteh et al., 2025; Pal et al., 2024) that depend on datasets created in the same setting. Given that our method may be adaptable to other compilable programming languages or other operating systems, we leave them for plausible future work. One of other potential drawbacks of our benchmark is the relatively small number of compilable repositories for the test set. However, we compensate for the quantity with intensive manual verification that produces ground-truth binary file and retrieval labels that facilitate both analysis and future developments. Although OSS-BUILD-AGENT shows competitive performance, we acknowledge the inherent instability of the agentic framework that may introduce variations in performance when reproducing the experiments. Also, we believe that agentic retrieval could be further enhanced with recent advancement of AI agent research, which ultimately improves the accuracy of the retrieval to enhance the overall compilation performance. We invite researchers to expand the potential of different agents' design philosophies and validate them on BUILD-BENCH to facilitate real-life developers and downstream research.

## 9 CONCLUSION

In this paper, we present a more challenging benchmark for building C and C++ source code repositories. Using BUILD-BENCH, we conducted a rigorous evaluation of different compilation methods, including our agentic baseline OSS-BUILD-AGENT. Analysis of module designs of agentic compilation methods pinpoints the challenging nature of the compilation task and shed lights on desirable approaches. We hope that our work contributes to the community by providing a suitable benchmark and inspires the community to build better agents for the task of OSS compilation.

## REPRODUCIBILITY STATEMENT

We take measures to ensure the reproducibility of this work. For dataset constructions, we include technical details of raw dataset we sample from and subsequent filtering in Section 2, and the sample size is justified in Section B. We provide the URLs to access both test set and validation set of BUILD-BENCH. For all experiment implementations, we submit the codebase along with manuscript.

## ETHICS STATEMENT

This work does not raise any ethical concerns and can be conducted without posing risks to individuals, communities, or the broader research environment.

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

## A    FILTERING KEYWORDS

A portion of keywords we used to filter out low-quality OSS including:

homework, assignment, tutorial, exercise, solution, course, student, university, college, class, lecture, demo, practice, presentation, getting started, hello world, starter code, sample code, example code, documentation.

## B    SAMPLE SIZE ESTIMATION

To estimate the minimum sample size required to measure a population proportion with 95% confidence and a margin of error of 5%, the standard formula for proportion estimation is:

$$n_0 = \frac{Z^2 p(1-p)}{E^2},$$ (1)

where $Z = 1.96$ for 95% confidence interval, $E = 0.05$ is the error margin, and $p = 0.5$ is chosen to maximize variance (i.e., yield the largest conservative sample size). This gives:

$$n_0 = \frac{1.96^2 \times 0.5 \times 0.5}{0.05^2} = 384.16.$$

For finite populations, we apply the finite population correction (FPC)Cochran (1977):

$$n = \frac{n_0}{1 + \frac{n_0 - 1}{N}} = \frac{384.16}{1 + \frac{383.16}{6,568,809}} \approx 384.14.$$ (2)

$$n = \frac{n_0}{1 + \frac{n_0 - 1}{N}} = \frac{384.16}{1 + \frac{383.16}{57,572}} \approx 384.14.$$ (3)

We round up to obtain a final required sample size of $n = 385$. We accordingly conduct a random sample of 385 repositories from the previously mentioned corpus to compose the final test set.

## C    LLM PROMPTS

### C.1    LLM-ASSISTED RETRIEVAL PROMPT

We use the following prompt to conduct LLM-Assisted Retrieval described in Section 3.1.

**Patch proposed by Agent**

You are an expert at extracting only the most relevant information (such as build instructions, dependency requirements) for building a repository from source for Linux based on provided documentation. Oftentimes, the steps of compilation or building from source for Linux should have already been included in the this file, otherwise, they may be stored in one other local files or external links. Please do not include anything that's not directly related to building from source like community support, License, Developer/API Documentation, Contribution, installation instructions for other use cases (such as Python etc.) or other irrelevant information.

Additionally, identify up to three external links and up to three internal links within the documentation, following the above objective.

For external URLs, extract the full url.
For internal file paths, use {readme_dir} as the base path to complete the partial paths. Besides, refine the internal paths to ensure they are valid paths. For example, the /docs/HowToGuides/GettingStarted.md#installing-dependencies should be completed as {readme_dir}/docs/HowToGuides/GettingStarted.md, as the section name after # would interfere with the validity of the path.

If you determine that there is no information directly related to building from source on Ubuntu like community support, License, Contribution, installation instructions for other use cases (such as Python etc.) or other irrelevant information, simply fill the field of "Build_Instructions" with "No build instructions found" but still try to find useful urls or internal links for External_URLs and Internal_Paths.

## C.2 LLM BASELINE PROMPTS

**Patch proposed by Agent**

You are an expert Linux build engineer working inside a
**Ubuntubased Docker container**. The pre-installed software
and libraries are as listed in the following dockerfile
content:
\{per\_installed\_libraries\_in\_docker\}

\#\#\# Your task
Generate a **sequence of Bash commands** (one command per
line, no comments, no explanations) that will:
1. Install every buildtime dependency needed to compile the
repository **{repo\_full\_name}** that lives at
**{repos\_dir\_in\_docker}**.
 Use noninteractive `apt-get update \&\& apt-get install -y `
when possible.
 Avoid PPAs unless strictly necessary.
 Assume you run as root, so no `sudo` is required.
2. **Detect build system and configure debug build:**
Examine the repository structure (files listed below) to
choose the proper build configuration command. Configure the
build system in Debug mode (i.e., include DWARF symbols,
disable optimizations).
3. **Install the main binary:**
Identify the primary or main binary (for example, the one
built from the projects main executable) and install it into
{repos\_dir\_in\_docker}
– Ensure the installation directory exists (create it if
necessary with `mkdir -p`).
– Copy the main binary into that directory and set executable
permissions if needed.

\#\#\# Strict requirements:
***Output only Bash commands, separated using the newline
character.**
Do not provide any explanations, markdown, or extra comments.
* The commands must be **fully sequential and ready-to-run**
when concatenated.
There should be no interactive prompts or assumptions beyond
what is provided.
* All steps must run successfully in a typical Docker Ubuntu
environment.
* Assume the current working directory is ** "/app" **.

\#\#\# Repository context
**Repo name:** {repo\_full\_name}
**Root path in container:** {repos\_dir\_in\_docker}

**README:**
{readme\_content}

**Toplevel file list:**
{files\_in\_root\_dir}

C.3 SYSTEM PROMPT FOR *Bash Command Generator*

Patch proposed by Agent

```
You are an helpful AI assistant that is an expert in
compiling cloned GitHub repositories and handling compilation
errors during the process by generating bash commands.
The current working directory is `/app`, and all commands
must use absolute paths referencing the repository's specific
clone directory, with no placeholders. The compilation
process runs inside a Docker container with root access, so
do not use `sudo`. Your suggested code must be complete and
executable, as the user cannot modify it. Ensure the target
repository is compiled with debug information, for instance
by adding `-g -O0` to compiler flags, and do not strip this
information after compilation. Whenever possible, use a
prefix or `DESTDIR` flag during the `make` command to save
compiled artifacts inside the clone directory. Always run
`make install` after compilation, using multiple cores to
speed up the process, but do not run `make check` or `make
test`. More detailed building instructions from the
repository will be provided, which you must follow. You
should attempt to fix any errors that occur. To end the
process upon success or failure, send a message explaining
the reason followed by the word "terminate," but never
include "terminate" in a response that also contains a code
block. Do not show appreciation in your responses; if "Thank
you" is said, reply only with "TERMINATE".
```

C.4 SYSTEM PROMPT FOR *Executor Agent*

**System Prompt**:
You are an AI assistant that can run bash commands or execute function calling and conduct the
process of GitHub repository compilation.

Table 4: Model sensitivity to base model choices.

| Model | Target URL Prediction Accuracy | | Target Trajectory Coverage (n=136) |
| | Overall (n=136) | Out-of-repo (n=18) | |
| --- | --- | --- | --- |
| GPT-4o | 73.8% | 5.6% | 81.2% |
| o3-mini | 72.8% | 5.6% | 81.0% |
| Gemini 2.5 Flash | 74.3% | 22.2% | 82.7% |
| Claude 3.7 Sonnet | 75.0% | 27.8% | 84.0% |
| Qwen3 235B | 74.3% | 22.2% | 82.5% |
| Qwen3 Coder 480B | 72.8% | 11.1% | 81.2% |

Table 5: Comparison of retrieval methods in terms of accuracy and retrieval cost per repository.

| Retrieval Method | ACC (Overall, N=136) | ACC (External, n=18) | Avg HTTP Requests / Repo | Avg Monetary Cost / Repo |
| --- | --- | --- | --- | --- |
| **TF-IDF** | 63.2% | N/A | N/A | N/A |
| **RAG** | 77.2% | 0.0% | 64.4 | ˜0.32M tokens (0.04 USD) *using text-embedding-3-large* |
| **LLM-Assisted Retrieval (Ours)** | 75.0% | 27.8% | Max 9 | ˜45K input + ˜780 output tokens (0.11 USD) using GPT-4o |

## D    RETRIEVAL BENCHMARK

We evaluate retrieval baselines on the secondary retrieval benchmark described in Section 6.2.

**Base model sensitivity**. In Table 4, we conduct additional experiments regarding our retrieval module, LLM-Assisted Retrieval, against different base models. Meanwhile, we also measure the retrieval trajectory coverage with the ground-truth labels, measuring how many intermediate urls have been captured by our retrieval modules. The results showcasing our retrieval module can capture around 80% of intermediate URLs or internal files, as well as the potentially useful build instructions contained in them.

**Retrieval strategies on retrieval benchmark**. We additionally implement a lightweight heuristic retrieval method TF-IDF as one of the retrieval baseline. The results are shown in Table 5.

For TF-IDF and RAG, we use a consistent URL-Hit@5 metric: a prediction is correct if the ground-truth instruction URL appears among the sources of the top-5 retrieved text snippets.

TF-IDF (in-repo documentation files only, since it is unable to explore external URLs) achieves 63.2%, whereas RAG baseline achieves 77.2%. Our LLM-assisted retriever (Claude 3.7 Sonnet) surpasses all methods on retrieval accuracy metric, and on the 18 repositories whose ground-truth is an external URL it reaches 27.8% versus 0% for RAG. These results suggest that lightweight heuristics (TF-IDF/RAG) are efficient and strong for in-repo instructions, whereas our LLM retriever substantially improves coverage on long-tail external instruction pages.

## E    TIME AND MONETARY COST

Here, we report the additional metrics that can be used to evaluate the cost of our methods. Results are shown in Table 6.

We use GPT-4o as reference for fair comparison with CompileAgent. On average of three runs, roughly 87K input tokens and 1.6K output tokens are consumed to compile each repository, combining at 0.23USD. Also, taking into account the retrieval cost, averaging in 0.11USD using GPT-4o, we are looking at around 0.34 USD per repo.

In terms of time consumption. Since all the experiments using OSS-BUILD-AGENT are conducted on the Kubernetes cluster, assigned 10 CPU cores per repository, and the baseline experiments are

Table 6: Comparison of time and API cost per repository across different compilation agents.

| Method | Time Cost per Repo (min) | LLM API Cost per Repo (USD) |
|---|---|---|
| OSS-BUILD-AGENT (GPT-4o) | 5.27 | $0.34 |
| CompileAgent (GPT-4o) | 6.70 | $0.16 |
| GHCC (Rule-Based) | 0.80 | 0 |
| Assemblage (Rule-Based) | N/A | 0 |

done in a single local server with default setting, we can only provide reference time consumption below. Assemblage does not release a reference time cost analysis in their manuscript.

We observed minor differences between the public implementation and the manuscript. We therefore report CompileAgent costs as reported in its manuscript. Moreover, GHCC shows minimal time consumption due to its one-shot execution strategyif the initial attempt fails, it terminates immediately without retrying.

## F    CASE STUDY 1: AGENTIC COMPILATION PATCHING SOURCE FILES

During our log analysis we observe that in some cases Agentic Compilation would attempt to fix the source files after encountring compilation errors and then continue building the project. *s9xie/hed* repository, part of BUILD-BENCHhas a code base that is 10 years old which relies on outdated packages and dependencies. It uses OpenCV v3 API calls and originally build to run on Ubuntu 14. Newer versions of OpenCV v4 updated their API, which causes this project to fail to build out-of-the-box on recent versions of Ubuntu. Based on error log that Agent received as part of feedback loop, it automatically patched the source files updating the occurrences of old API and successfully compiled the repository. For instance, it updated CV_LOAD_IMAGE_COLOR to IMREAD_COLOR. It showcases the potential of AI-based compilation method of patching deemed 'uncompilable' repositories, whereas a rule-based approach would never be able to fix it automatically without human-assistance.

**Error Log**

```
/app/k8s_compiled_repos/hed/src/caffe/layers/window_data_lay↵
↪   er.cpp: In member function virtual void
↪   caffe::WindowDataLayer<Dtype>::load_batch(caffe::Batch<D↵
↪   type>*):
/app/k8s_compiled_repos/hed/src/caffe/layers/window_data_lay↵
↪   er.cpp:288:42: error: CV_LOAD_IMAGE_COLOR was not
↪   declared in this scope
  288 |        cv_img = cv::imread(image.first,
    ↪   CV_LOAD_IMAGE_COLOR);
      |
        ↪   ^~~~~~~~~~~~~~~~~~~~
...
...
...
/app/k8s_compiled_repos/hed/src/caffe/util/io.cpp: In
↪   function cv::Mat caffe::ReadImageToCVMat(const string&,
↪   int, int, bool):
/app/k8s_compiled_repos/hed/src/caffe/util/io.cpp:73:34:
↪   error: CV_LOAD_IMAGE_COLOR was not declared in this scope
   73 |   int cv_read_flag = (is_color ? CV_LOAD_IMAGE_COLOR
    ↪   :
      |                                  ^~~~~~~~~~~~~~~~~~~~
/app/k8s_compiled_repos/hed/src/caffe/util/io.cpp:74:5:
↪   error: CV_LOAD_IMAGE_GRAYSCALE was not declared in this
↪   scope
   74 |      CV_LOAD_IMAGE_GRAYSCALE);
      |      ^~~~~~~~~~~~~~~~~~~~~~~~
...
...
/app/k8s_compiled_repos/hed/src/caffe/layers/window_data_lay↵
↪   er.cpp: In member function virtual void
↪   caffe::WindowDataLayer<Dtype>::load_batch(caffe::Batch<D↵
↪   type>*):
```

**Patch proposed by Agent**

```
Agent Compilation :

```bash
# Patch the OpenCV constants to use new names              \
↪   (IMREAD_COLOR and IMREAD_GRAYSCALE) in the affected     \
↪   source files.
sed -i 's/CV_LOAD_IMAGE_COLOR/IMREAD_COLOR/g'              \
↪   /app/k8s_compiled_repos/hed/src/caffe/util/io.cpp
sed -i 's/CV_LOAD_IMAGE_GRAYSCALE/IMREAD_GRAYSCALE/g'      \
↪   /app/k8s_compiled_repos/hed/src/caffe/util/io.cpp
sed -i 's/CV_LOAD_IMAGE_COLOR/IMREAD_COLOR/g'             \
↪   /app/k8s_compiled_repos/hed/src/caffe/layers/window_↵    \
↪   data_layer.cpp
```
```

## G  CASE STUDY 2: CMAKE ERROR DUE TO DIRECTORY CHANGES

Agents fail to compile repositories due to various reasons, as described in 6.3. For instance, when compiling GitHub repository *bernhard-schmitzer/optimal-transport*, the agent invoked cmake from the wrong working directory; further troubleshooting would have revealed that the CMakeLists.txt resides in the src subdirectory. While changing to a different directory is a simple and straghtforward solution, but the agent did not perform sufficient troubleshooting.

---

**Compilation commands**

```bash
cd /app/k8s_compiled_repos/optimal-transport/v0.2.0
mkdir -p build && cd build
cmake -DCMAKE_BUILD_TYPE=Debug -DCMAKE_CXX_FLAGS="-g -O0"  \
↪  -DCMAKE_C_FLAGS="-g -O0" ..
make -j15
make install                                               \
↪  DESTDIR=/app/k8s_compiled_repos/optimal-transport
```

---

**Execution log**

```
exitcode: 2 (execution failed)
CMake Error: The source directory                          \
↪  "/app/k8s_compiled_repos/optimal-transport/v0.2.0"      \
↪  does not appear to contain CMakeLists.txt.
```

---

## H  CASE STUDY 3: INSUFFICIENT TROUBLESHOOTING

Agents fail to compile repositories due to various reasons, as described in 6.3. For instance, when compiling GitHub repository *blitz3d-ng/package* , the agent attempts to first initialize the Git submodules and the execute make to compile the repository. However, the execution failed. First, the current user running the git submodule update command is not the owner of the specified repository directory, causing *dubious ownership* error. Then this results in a follow-up error in make that indicates *No rule to make target 'install'*.

To resolve the error, the agent must address the **submodule initialization error**, as it is the **root cause**. If the agent focuses only on tackling the follow-up make error, the resolution will ultimately fail, highlighting the need for agents to diagnose and fix the primary issue in the execution chain.

**Compilation commands**

```bash
apt-get update
apt-get install -y git ninja-build libxml2-dev zlib1g-dev   \
 ↪  build-essential autoconf libtool gettext autopoint       \
 ↪  gperf cmake clang libwxgtk3.0-gtk3-dev libxrandr-dev      \
 ↪  libxinerama-dev libxcursor-dev uuid-dev                   \
 ↪  libfontconfig1-dev
cd /app/compiled_repos/package
git submodule update --init --recursive
source /app/k8s_compiled_repos/package/env.sh
make -j15 llvm
make -j15 CPPFLAGS='-g -O0' CXXFLAGS='-g -O0'
make install PREFIX=/app/compiled_repos/package
```

**Execution log**

```
exitcode: 2 (execution failed)
git submodule update failed:
fatal: detected dubious ownership in repository at        \
 ↪  '/app/k8s_compiled_repos/package'

...

make install failed: No rule to make target 'install'
```

## I   USE OF LLMS

In this submission, we use LLM to polish writing and correct grammar mistakes. No additional usage is conducted.

## J   INFORMATION ABOUT REPOSITORIES

Table 7: The composition of BuildBench

| Project | Topic | Existing Guide | | No Guide |
| --- | --- | --- | --- | --- |
| | | InRepo | NotInRepo | |
| LithiumX | gaming | ✓ | × | × |
| virgil-crypto | crypto | ✓ | × | × |
| cqmetrics | metrics | ✓ | × | × |
| infact | c-plus-plus | × | ✓ | × |
| OpenOCD-Nuvoton | networking | ✓ | × | × |
| freesasa | bioinformatics | ✓ | × | × |
| librtmp | networking | × | × | ✓ |
| stderred | cli | ✓ | × | × |
| dhewm3 | gaming | ✓ | × | × |
| LISP | interpreter | ✓ | × | × |
| hostap-wpa3 | security | ✓ | × | × |
| pure-lang | functional-programming | × | ✓ | × |
| andvaranaut | gaming | ✓ | × | × |
| srs | networking | × | ✓ | × |
| fsarchiver | linux | × | ✓ | × |
| PGE | gaming | ✓ | × | × |
| is_jsonb_valid | database | ✓ | × | × |
| bimpy | gui | ✓ | × | × |
| leopard | database | × | × | ✓ |
| rvmparser | parser | ✓ | × | × |
| android-performance | android | × | ✓ | × |
| MaslOS | kernel | ✓ | × | × |
| megaglest-source | gaming | ✓ | × | × |
| sharebox-fs | filesystem | × | × | ✓ |
| mclinker | c-plus-plus | ✓ | × | × |
| patcher9x | patch | ✓ | × | × |
| berkeley-softfloat-3 | c | × | ✓ | × |
| spacenavd | driver | ✓ | × | × |
| AmBinaryEditor | android | × | × | ✓ |
| mm3d | 3d-model | ✓ | × | × |
| ULTRA | algorithms | ✓ | × | × |
| gdal | raster | × | ✓ | × |
| xprompt | x11 | ✓ | × | × |
| Proxifier-For-Linux | proxifier | ✓ | × | × |
| luaevent | bindings | ✓ | × | × |
| ggmorse | remote-sensing | ✓ | × | × |
| SourceAutoRecord | gameing | ✓ | × | × |
| oscam-patched-old | softcam | ✓ | × | × |
| DupGen_finder | bioinformatics | ✓ | × | × |
| xz | c | ✓ | × | × |
| openat | finance | ✓ | × | × |
| sysuh3c | networking | ✓ | × | × |
| nOS | microcontrollers | × | ✓ | × |
| lightweight-crypto | crypto | ✓ | × | × |
| telegram-bot-api | telegram | ✓ | × | × |
| jsonsl | networking | ✓ | × | × |
| trocksdb | database | ✓ | × | × |
| pgmagick | c-plus-plus | ✓ | × | × |
| vkcube | c | × | × | ✓ |
| SignalRecovery | c | ✓ | × | × |
| rover | file-browser | ✓ | × | × |
| EasyRTSPLive | rtsp | ✓ | × | × |
| pysubnettree | networking | ✓ | × | × |
| poster | operating-system | ✓ | × | × |
| Multi_Sensor_Fusion | graphics | ✓ | × | × |
| web-server | multi-threading | ✓ | × | × |
| q4wine | wine | ✓ | × | × |
| libsrt | c | ✓ | × | × |
| BundleFusion_Ubuntu_Pangolin | 3d-model | ✓ | × | × |
| iotkit-embedded | iot | × | ✓ | × |
| semu | emulator | ✓ | × | × |
| level-ip | networking | ✓ | × | × |
| NovaLSM | database | ✓ | × | × |
| package | 3d-model | ✓ | × | × |
| DuckX | c-plus-plus | ✓ | × | × |
| Brayns | graphics | ✓ | × | × |
| PLADE | c-plus-plus | ✓ | × | × |
| spot-on | c-plus-plus | ✓ | × | × |
| air | php | ✓ | × | × |
| cam2web | graphics | ✓ | × | × |
| twemproxy | networking | ✓ | × | × |
| SNANDer | hardware | ✓ | × | × |
| aocla | c | ✓ | × | × |
| obs-gnome-screencast | linux | × | × | ✓ |
| Dwarf-Therapist | gaming | × | ✓ | × |

Table 8: The composition of BuildBench

| Project | Topic | Existing Guide | | No Guide |
|---------|-------|------|------|---------|
| | | InRepo | NotInRepo | |
| yuv2rgb | c | ✓ | × | × |
| ssocks | networking | ✓ | × | × |
| myMPD | audio | × | ✓ | × |
| ircDDBGateway | networking | ✓ | × | × |
| Doom8088 | gaming | ✓ | × | × |
| Lampray | gaming | ✓ | × | × |
| Pangolin | 3d-model | ✓ | × | × |
| mercury | hpc | × | ✓ | × |
| navmesh | c-plus-plus | ✓ | × | × |
| nginx-http-shibboleth | nginx | × | ✓ | × |
| cavif | graphics | ✓ | × | × |
| TexasSolver | gaming | × | × | ✓ |
| prefix | database | ✓ | × | × |
| file | c | ✓ | × | × |
| realm-core | database | ✓ | × | × |
| redis-leveldb | caching | ✓ | × | × |
| tbox | networking | ✓ | × | × |
| parrot | virtual-machine | ✓ | × | × |
| mpio | c-plus-plus | × | × | ✓ |
| G1fitting | c-plus-plus | ✓ | × | × |
| libxd | graphics | × | ✓ | × |
| webserver | networking | × | × | ✓ |
| TinyBIOS | operating-system | ✓ | × | × |
| quartz | emulator | ✓ | × | × |
| csldr | gaming | × | × | ✓ |
| nap | graphics | × | ✓ | × |
| pysqlite3 | database | ✓ | × | × |
| OpenGL-Renderer | graphics | ✓ | × | × |
| shared | c-plus-plus | ✓ | × | × |
| thot | machine-learning | × | ✓ | × |
| SaBRe | binary-rewriting | ✓ | × | × |
| renderdoc | graphics | ✓ | × | × |
| that_editor | editor | ✓ | × | × |
| simdjson_php | php | ✓ | × | × |
| libRaptorQ | c-plus-plus | ✓ | × | × |
| QHotkey | cross-platform | ✓ | × | × |
| PIMSim | emulator | ✓ | × | × |
| nanobind | python | × | × | ✓ |
| tracer | graphics | ✓ | × | × |
| gtsa | gaming | ✓ | × | × |
| dplus-c | android | × | × | ✓ |
| otherside | virtual-machine | ✓ | × | × |
| libmolgrid | machine-learning | ✓ | × | × |
| obsqr | android | ✓ | × | × |
| fftocean | c-plus-plus | ✓ | × | × |
| HOLLOW | c | ✓ | × | × |
| dbcc | c | ✓ | × | × |
| rocketmq-client-cpp | rocketmq | ✓ | × | × |
| MNN | machine-learning | × | ✓ | × |
| pepecoin | crypto | ✓ | × | × |
| bxxt | c | ✓ | × | × |
| Quartic | c-plus-plus | ✓ | × | × |
| csol | gaming | ✓ | × | × |
| cli-gpt | machine-learning | ✓ | × | × |
| eekf | c | ✓ | × | × |
| mx | c | ✓ | × | × |
| atto | editor | ✓ | × | × |
| lucy | attic | ✓ | × | × |
| uvConvertor | c-plus-plus | ✓ | × | × |
| WendzelNNTPd | database | ✓ | × | × |
| vxsig | antivirus | ✓ | × | × |

Table 9: The composition of BuildBench

| Project | Topic | Existing Guide | | No Guide |
|---|---|---|---|---|
| | | InRepo | NotInRepo | |
| pg_amqp | database | ✓ | × | × |
| RtspServer | networking | ✓ | × | × |
| ctypes.sh | editor | ✓ | × | × |
| Turbo-Base64 | crypto | ✓ | × | × |
| berserk | gaming | ✓ | × | × |
| TheWiggler | c-plus-plus | ✓ | × | × |
| poedit | translation | ✓ | × | × |
| fastq-tools | c | ✓ | × | × |
| munge | hpc | × | ✓ | × |
| ChowDSP-VCV | networking | ✓ | × | × |
| sxhkd | c | × | × | ✓ |
| GuiLite | c-plus-plus | ✓ | × | × |
| pslab-firmware | hardware | ✓ | × | × |

