# OpenReview forum: "Build-Bench: Benchmarking LLM Agents on Compiling Real-World Open Source Software"
_ICLR.cc/2026/Conference — Submitted to ICLR 2026_

### Official Review · Reviewer_atPJ · 2025-10-27

**Soundness:** 3
**Presentation:** 3
**Contribution:** 2
**Rating:** 4
**Confidence:** 3

**Summary:**

This paper presents Build-Bench, a benchmark on compiling open-source software project with LLM Agents. The authors propose more realistic challenges of OSS compilation, mainly targeting github repositories of C/C++ projects. The authors also propose a strong baseline for LLM agent which enhance build instructions based on a retrieval module.

**Strengths:**

(1) The authors too greate efforts to enhance the data selection process for collecting Build-Bench problems. Especially the distribution of stargazer counts more aligns with data collected from the wild on github. Furthermore the dataset demonstrate a wide variety of build systems and tool chains.

(2) A multi-agent LLM agentic baseline is proposed to solve Build-Bench. The first stage LLM are used to extract relevant instructions and access relevant links and files to generate compilation instruction. The second stage consists of a flow-based method, where a bash command generator compiles bash commands and the executor agent executes the command.

**Weaknesses:**

(1) Marginal improvements over CompileAgent. It seems that CompileAgent with Retrieval on GPT-4o works quite well, and only a marginal improvement on flexible validated successes is presented.

(2) Success metric is determined by compilation and existence of binary links, which seems too tailored to C/C++.

**Questions:**

(1) How would the success method be extended to programming languages outsied of C/C++. Is using unit tests etc. a more rigourous method for evaluation?

(2) Why solely focus on C/C++? Any plans to extend to other programming languages (such as Python etc.)?

---

> ### Author Response · Authors · 2025-11-23
> **Response to Reviewer atPJ**
>
> Thank you for recognizing our contribution to the robust data selection process and agentic compilation methods. Please find our response below.
>
> >__Weakness 1__: Marginal improvements over CompileAgent. It seems that CompileAgent with Retrieval on GPT-4o works quite well, and only a marginal improvement on flexible validated successes is presented.
>
> __A1__: Our main contribution is the creation of a more realistic and challenging benchmark dataset for compilation tasks, along with various analysis and evaluation of agentic build methods.
>
> Our proposed agent, while achieving a marginal performance improvement over CompileAgent, still increases the performance on our benchmark, showcasing that a simple self-correction agentic workflow can outperform complex multi-agent systems like CompileAgent on compilation tasks.
>
> ---
> >__Weakness 2__: The success metric is determined by compilation and existence of binary links, which seems too tailored to C/C++. __Question 1__: How would the success method be extended to programming languages outside of C/C++. Is using unit tests etc. a more rigorous method for evaluation?
>
> __A2__: We adopt the existence of binary files (completion check) and the exact matching of generated binary names with ground-truth labels (flexible/strict validated success) as the core evaluation metrics. This approach provides a rigorous and automated means to confirm that the compilation process completes successfully, while still providing coverage for both partial and fully correct builds.
>
> Linux is the most common operating system used by many works to create binary datasets by compiling C/C++ repositories as well as research on downstream tasks  [1-6], and our work’s main motivation is to create a benchmark and agentic compilation methods to automate the compilation task. Extending this work to other operating systems and programming languages is a plausible future work, and we have updated our Limitation section (Section 8) accordingly.
>
> Lastly, unit tests can indeed be an effective supplement to our evaluation method, as it also evaluates functionality validity after compilation. However, we found very few repositories in our benchmark containing unit tests. Many projects in BUILD-BENCH are small-scale or personal repositories without testing suites, which restrained us from integrating functionality checks. Nevertheless, we believe unit tests could enhance the robustness of evaluation when available.
>
> ---
> >__Question 2__: Why solely focus on C/C++? Any plans to extend to other programming languages (such as Python etc.)?
>
> __A3__: Please attend to our responses to Question 1 and Weakness 2.
>
> ---
> __References__:
> 1. Jeremy Lacomis, Pengcheng Yin, Edward J. Schwartz, Miltiadis Allamanis, Claire Le Goues, Graham Neubig, and Bogdan Vasilescu. 2020. DIRE: a neural approach to decompiled identifier naming. In Proceedings of the 34th IEEE/ACM International Conference on Automated Software Engineering (ASE '19).
> 2. K. K. Pal et al., ""Len or index or count, anything but v1": Predicting Variable Names in Decompilation Output with Transfer Learning," 2024 IEEE Symposium on Security and Privacy (SP), San Francisco, CA, USA, 2024, pp. 4069-4087.
> 3. Danning Xie, Zhuo Zhang, Nan Jiang, Xiangzhe Xu, Lin Tan, and Xiangyu Zhang. 2024. ReSym: Harnessing LLMs to Recover Variable and Data Structure Symbols from Stripped Binaries. In Proceedings of the 2024 on ACM SIGSAC Conference on Computer and Communications Security (CCS '24).
> 4. Hu, P., Liang, R., & Chen, K. (2024). DeGPT: Optimizing Decompiler Output with LLM. Proceedings 2024 Network and Distributed System Security Symposium.
> 5. Clemens-Alexander Brust, Tim Sonnekalb, and Bernd Gruner. 2023. ROMEO: A binary vulnerability detection dataset for exploring Juliet through the lens of assembly language. Comput. Secur. 128, C (May 2023).
> 6. Clemens-Alexander Brust, Tim Sonnekalb, and Bernd Gruner. 2023. ROMEO: A binary vulnerability detection dataset for exploring Juliet through the lens of assembly language. Comput. Secur. 128, C (May 2023).

---

> ### Author Response · Authors · 2025-11-28
>
> I hope this message finds you well. As the discussion period is ending in less than a week. I wanted to ensure we have addressed all your concerns fully. If there are any points or feedback you'd like us to address, please let us know. Your input and feedback are valuable to us to improve our work. We appreciate your time and effort in reviewing our paper.

---

### Official Review · Reviewer_Z2nX · 2025-10-30

**Soundness:** 3
**Presentation:** 3
**Contribution:** 3
**Rating:** 6
**Confidence:** 3

**Summary:**

This paper presents BUILD-BENCH, a benchmark dataset derived from 385 randomly sampled C/C++ GitHub repositories, with 148 manually validated as compilable and annotated with ground-truth binary artifacts and documentation URLs. The authors propose OSS-BUILD-AGENT, a multi-agent LLM system featuring an LLM-assisted retrieval module and iterative error resolution. Evaluations against rule-based baselines (GHCC, Assemblage), single-turn LLM approaches, and CompileAgent using multiple frontier models demonstrate that the best configuration achieves 66.4% strict validated success rate, substantially outperforming existing methods on this more challenging and representative benchmark.

**Strengths:**

BUILD-BENCH employs principled random sampling from 6.57M repositories, addressing the selection bias in COMPILEAGENTBENCH where 99.88% of real-world C/C++ projects have fewer than 500 stars. The benchmark's build system diversity (Make, CMake, Autotools, MSBuild, custom scripts) and ground-truth annotations enable rigorous evaluation. The experimental design is comprehensive, spanning multiple baselines and frontier LLMs (GPT-4o, o3-mini, Claude 3.7-Sonnet, Gemini 2.5, Qwen3), with transparency regarding stochasticity through repeated runs and pass@k analysis. The retrieval module comparison with CompileAgent provides actionable design insights, particularly regarding documentation-first traversal versus build-script-focused approaches.

**Weaknesses:**

- The benchmark's scope is restricted to Linux C/C++ projects, with no evaluation on Windows, macOS, or mixed-language codebases, limiting generalizability claims.
- Manual exclusion of "uncompilable" projects (criteria a-d in Section 2) may introduce selection bias toward better-documented repositories; the paper would benefit from publishing exclusion statistics and considering a "hard mode" supplemental dataset.
- Critical practical metrics are absent: per-repository runtime, API token consumption, and monetary costs are essential for practitioners evaluating adoption trade-offs.
- Ablation studies are insufficient—the paper does not isolate retrieval quality versus agent architecture, evaluate lightweight retrieval heuristics, or test retrieval module sensitivity to base LLM choice.

**Questions:**

1. Can the authors provide comprehensive efficiency metrics—average runtime per repository, total API calls, token consumption, and estimated costs—for OSS-BUILD-AGENT compared to CompileAgent and rule-based baselines?
2. What are the exclusion statistics for the 237 removed repositories? Specifically, what percentages were excluded due to cross-platform requirements, missing dependencies, broken builds, or other criteria?
3. Have you conducted ablation studies on the retrieval module: (a) performance with different base LLMs, (b) comparison with lightweight heuristics (regex-based link extraction, TF-IDF), and (c) impact of iteration depth and link count limits?

---

> ### Author Response · Authors · 2025-11-23
> **Response to Reviewer Z2nX (1/2)**
>
> We thank the reviewer for the careful reading and constructive suggestions. Below, we address each point and specify the changes we made to the manuscript.
>
> ---
> >__Weakness 1__: The benchmark's scope is restricted to Linux C/C++ projects, with no evaluation on Windows, macOS, or mixed-language codebases, limiting generalizability claims.
>
> __A1__: Linux is the operating system used by many related works that create binary datasets from building C/C++ projects as well as applying these binaries to downstream tasks [1-6]. Our work’s main motivation is creating a benchmark and agentic compilation methods to automate such a task. Extending this work to other operating systems and programming languages is a future effort, and we updated our Limitation section (Section 8) accordingly.
>
> ---
> >__Weakness 2__: Manual exclusion of "uncompilable" projects (criteria a-d in Section 2) may introduce selection bias toward better-documented repositories; the paper would benefit from publishing exclusion statistics and considering a "hard mode" supplemental dataset. __Question2__: What are the exclusion statistics for the 237 removed repositories? Specifically, what percentages were excluded due to cross-platform requirements, missing dependencies, broken builds, or other criteria?
>
>
> __A2__: We acknowledge that human labeling may introduce a selection bias toward well-documented repositories, so we also explicitly instruct the human experts to conduct best effort compilation to repositories that do not have documentation. As a result, the test set still includes 13 repositories (9%) without build instructions that were verified as compilable, ensuring broader coverage and mitigating this bias.
>
> During labeling, human experts excluded all uncompilable repositories. To define the exclusion criteria, we surveyed around 100 projects to document the common reasons experts failed to compile them.
> | Reason                               | Count |
> |--------------------------------------|--------|
> | Broken code requiring significant changes | 30     |
> | Windows / other cross-platforms      | 27     |
> | Missing or broken dependencies       | 19     |
> | Tutorials / classwork                | 8      |
> | Header files only                    | 6      |
> | Non C/CPP                            | 3      |
> | Missing build instructions           | 1      |
> | Required Login                       | 1      |
>
> We did not include the uncompilable repositories during experiments, but they are still released as part of Build-Bench, as we hope that future, more powerful models will be able to compile these repositories, even in cases that require substantial codebase changes.
>
> ---
> >__Weakness 3__:  Critical practical metrics are absent: per-repository runtime, API token consumption, and monetary costs are essential for practitioners evaluating adoption trade-offs. __Question 1__: Can the authors provide comprehensive efficiency metrics—average runtime per repository, total API calls, token consumption, and estimated costs—for OSS-BUILD-AGENT compared to CompileAgent and rule-based baselines?
>
>
> __A3__: We report the additional metrics that can be used to evaluate the adoption cost for developers.
>
> Adding together the cost of retrieval and compilation stages of OSS-Build-Agent using GPT-4o, the approximation of time and monetary cost is 5.27 minutes and $0.34 per repository.
> | System / Agent           | Time Cost Per Repo (Min) | LLM API Cost Per Repo (USD) |
> |---------------------------|--------------------------|------------------------------|
> | OSS-BUILD-AGENT - GPT-4o  | 5.27                     | $0.34                        |
> | CompileAgent - GPT-4o     | 6.7                      | $0.16                        |
> | GHCC                      | 0.8                      | $0 (Rule-Based)              |
> | Assemblage                | N/A                      | $0 (Rule-Based)              |
>
> Full breakdown of the cost can be found in Appendix E.

---

> ### Author Response · Authors · 2025-11-23
> **Response to Reviewer Z2nX (2/2)**
>
> > __Weakness 4__: Ablation studies are insufficient—the paper does not isolate retrieval quality versus agent architecture, evaluate lightweight retrieval heuristics, or test retrieval module sensitivity to base LLM choice. __Question3__: Have you conducted ablation studies on the retrieval module: (a) performance with different base LLMs, (b) comparison with lightweight heuristics (regex-based link extraction, TF-IDF), and (c) impact of iteration depth and link count limits?
>
> __A4__: For clarity, we respond to Weakness 4 and Question 3 together.
>
> 1. _Ablation on retrieval quality with different retrieval strategies._
>
> We added both RAG and TF-IDF as retrieval baselines. We show below the results of evaluating them on the secondary retrieval benchmark (described in Section 6.2). These results suggest that lightweight heuristics (TF-IDF/RAG) are efficient and strong for in-repo instructions, whereas our LLM retriever substantially improves coverage on long-tail external instruction pages.
>
> More details about the implementation of RAG and different retrieval strategies’ influence on the compilation task can be found in line 414 to 430. The implementation of TF-IDF can be found in Appendix Section D.
>
> | Retrieval Method               | ACC (Overall, N=136) | ACC (External, n=18) | Cost (Avg HTTP Requests Per Repo) | Cost (Avg Monetary Per Repo) |
> |---------------------------------|----------------------|----------------------|-----------------------------------|-------------------------------|
> | TF-IDF                          | 63.2%                | N/A                  | N/A                               | N/A                           |
> | RAG                             | 77.2%                | 0.0%                 | 64.4                              | ~0.32M tokens → $0.04 USD (using text-embedding-3-large) |
> | LLM-Assisted Retrieval (Ours)   | 75.0%                | 27.8%                | Max 9                             | ~45K input + ~780 output tokens → / $0.11 USD (GPT-4o)
> ---
> 2. _Ablation on compilation task performance with different retrieval strategies._
>
> To isolate retrieval quality vs agent architecture, we tested RAG, LLM-Assisted Retrieval, No Retrieval and Perfect Retrieval on the end-to-end compilation task, and the results are provided in Table 3 in Section 6.2.
>
> | Retrieval Methods (w/ OSS-BUILD-AGENT) | Unvalidated (%) | Validated Success (%) - Strict | Validated Success (%) - Flexible |
> |----------------------------------------|-----------------|--------------------------------|----------------------------------|
> | No Retrieval (Baseline)                | 67.6            | 48.0                           | 50.7                             |
> | RAG (Ablation)                         | 76.4 (+8.8)     | 58.8 (+10.8)                   | 64.9 (+14.2)                     |
> | LLM-Assisted Retrieval                 | 79.9 (+12.3)    | 63.1 (+15.1)                   | 68.5 (+17.8)                     |
> | Perfect Retrieval (Ablation)           | 79.1 (+11.5)    | 68.2 (+20.2)                   | 70.9 (+20.2)                     |
>
> In short, Perfect Retrieval sets the upper bound on validated correctness. It suggests that accurate retrieval of that human-written build instructions can drastically improve performance. Our proposed LLM-Assisted Retrieval consistently outperforms RAG in all metrics by a significant margin, with 4.3% more in the strict validated success percentage.
>
> ---
> 3. _Ablation on model sensitivity._
>
> We conducted retrieval experiments with different base models to test the retrieval module’s model sensitivity. Overall, the retrieval performance of models is consistent with their compilation performance, and the retrieval performance also scales with model’s intelligence. We note even the best performing model, Claude 3.7 Sonnet, can only achieve 27.8%  performance when handling out-of-repo targets, indicating the future needs of improving retrieval performance of compilation tasks. We have included the results in Table 4 and the analysis in Appendix Section D.
>
> ---
> 4. _Ablation on retrieval depth._
>
> We chose depth=3 for our retrieval module, because the average length of trajectories (path to build instruction url) is 2.56. To consider the performance drop of using a smaller retrieval depth, we additionally conducted an retrieval experiment for depth=2 using GPT-4o. The retrieval accuracy dropped from 73.8% to 72.1%, suggesting that the depth of retrieval does contribute to the overall retrieval quality of our retrieval module.

---

> ### Author Response · Authors · 2025-11-23
> **References**
>
> 1. Jeremy Lacomis, Pengcheng Yin, Edward J. Schwartz, Miltiadis Allamanis, Claire Le Goues, Graham Neubig, and Bogdan Vasilescu. 2020. DIRE: a neural approach to decompiled identifier naming. In Proceedings of the 34th IEEE/ACM International Conference on Automated Software Engineering (ASE '19).
> 2. K. K. Pal et al., ""Len or index or count, anything but v1": Predicting Variable Names in Decompilation Output with Transfer Learning," 2024 IEEE Symposium on Security and Privacy (SP), San Francisco, CA, USA, 2024, pp. 4069-4087.
> 3. Danning Xie, Zhuo Zhang, Nan Jiang, Xiangzhe Xu, Lin Tan, and Xiangyu Zhang. 2024. ReSym: Harnessing LLMs to Recover Variable and Data Structure Symbols from Stripped Binaries. In Proceedings of the 2024 on ACM SIGSAC Conference on Computer and Communications Security (CCS '24).
> 4. Hu, P., Liang, R., & Chen, K. (2024). DeGPT: Optimizing Decompiler Output with LLM. Proceedings 2024 Network and Distributed System Security Symposium.
> 5. Clemens-Alexander Brust, Tim Sonnekalb, and Bernd Gruner. 2023. ROMEO: A binary vulnerability detection dataset for exploring Juliet through the lens of assembly language. Comput. Secur. 128, C (May 2023).
> 6. Clemens-Alexander Brust, Tim Sonnekalb, and Bernd Gruner. 2023. ROMEO: A binary vulnerability detection dataset for exploring Juliet through the lens of assembly language. Comput. Secur. 128, C (May 2023).

---

> ### Author Response · Authors · 2025-11-28
>
> We hope this message finds you well. As the discussion period concludes soon, we wanted to confirm that we’ve addressed all of your concerns. If you have any additional questions or feedback, please feel free to let us know. We appreciate your thoughtful review and the time you’ve put into our work.

---

### Official Review · Reviewer_tMyd · 2025-10-31

**Soundness:** 3
**Presentation:** 2
**Contribution:** 2
**Rating:** 4
**Confidence:** 3

**Summary:**

This paper presents Build-Bench, a benchmark designed to evaluate LLMs on software build and compilation tasks. The benchmark collects real open-source projects with diverse build systems and automatically generates build tasks that simulate dependency errors, configuration failures, and compilation issues. Each task provides the model with build logs, dependency information, and target build commands as input, requiring it to output repair actions such as command corrections, dependency installations, or configuration updates. Tasks are automatically evaluated via sandbox compilation. Experiments on multiple LLMs show that while current models can handle simple dependency errors, they struggle with complex build environments and cross-language configurations.

**Strengths:**

The benchmark extends the evaluation of LLMs from code understanding to practical software build and dependency repair tasks. Its design is systematic and well-documented, using reproducible environments and real-world build systems. The task formulation is clear and grounded in realistic developer workflows. This benchmark provides insights into the limitations of current models in dependency management, configuration understanding, and handling multi-language projects.

**Weaknesses:**

1. Although the projects are sourced from real open-source repositories, the build failures are synthetically generated through automated error injection, which may reduce their realism. In actual development scenarios, build failures often arise from more complex and interdependent causes that maybe not fully represented in the benchmark.
2. The experimental analysis lacks depth, with limited discussion of error categories or their influence on model behavior. A more detailed breakdown of failure types probably can reveal additional insights and lead to stronger conclusions.
3. The evaluation metrics are relatively narrow and strictly binary, while build repair is inherently a gradual process. Fully successful repairs may also differ in their complexity or practical soundness, which the current evaluation framework does not capture.

**Questions:**

See weakness please

---

> ### Author Response · Authors · 2025-11-23
> **Response to Reviewer tMyd**
>
> We appreciate the reviewer’s time and effort in providing us with helpful feedback. We hereby prepared a point-to-point clarification to address the reviewer’s concern.
>
> ---
> > __Weakness 1__: Although the projects are sourced from real open-source repositories, the build failures are synthetically generated through automated error injection, which may reduce their realism. In actual development scenarios, build failures often arise from more complex and interdependent causes that may not fully be represented in the benchmark.
>
>
> __A1__: For clarity, Build-Bench does _not_ inject synthetic build errors. All repositories in Build-Bench are kept untouched after cloning from GitHub, and we did not intentionally corrupt them before building. The build failures are produced when the generated bash commands by build methods are executed, and they are feedbacks from the Linux environment. These methods never synthetically generate any build failures.
>
> We added a sentence at line 137-140 to clarify that all repositories used in experiments are cloned as-is and intact to avoid future confusion.
>
>
> ---
> > __Weakness 2__: The experimental analysis lacks depth, with limited discussion of error categories or their influence on model behavior. A more detailed breakdown of failure types probably can reveal additional insights and lead to stronger conclusions.
>
> __A2__: We thank the reviewer for their feedback. We have two case studies (Section G and H in Appendix) for the insufficient error resolution failure mode in the previous manuscript.  We added additional analysis in Section 6.3 accordingly, specifically we provide more details and insights of the root causes and agentic behavior regarding the three leading failure modes. We hope this can strengthen the depth of experimental analysis and the value of our work.
>
> ---
> > __Weakness 3__: The evaluation metrics are relatively narrow and strictly binary, while build repair is inherently a gradual process. Fully successful repairs may also differ in their complexity or practical soundness, which the current evaluation framework does not capture.
>
>
> __A3__: We agree that the build repair is gradual. We also considered using dynamic evaluation metrics: calculating the percentage overlaps between functions presented in the source code and compiled binary files, but it is much less rigorous and also suffers from a high false-positive rate and difficulty to establish a reasonable percentage threshold.
>
> This is why we use strict/flexible validated successes derived from labeled ground truth to reduce false positives from partial/irrelevant outputs, instead of a naive completion check. It is a more robust and deterministic way of checking building success.
>
> Moreover, we also evaluate through pass@K and the number of error-resolution attempts, which shows improvements with more retries (Table 2 and Figure 4 in the updated manuscript).

---

> ### Author Response · Authors · 2025-11-28
>
> I hope this message finds you well. With the discussion period ending in less than a week, we wanted to check in to make sure we’ve fully addressed your concerns. If there are any remaining points or feedback you’d like us to respond to, please let us know. Your input is invaluable in helping us improve the paper, and we greatly appreciate the time and care you’ve put into the review.

---

### Official Review · Reviewer_y24p · 2025-10-31

**Soundness:** 2
**Presentation:** 2
**Contribution:** 2
**Rating:** 4
**Confidence:** 3

**Summary:**

The paper BUILD-BENCH introduces a new benchmark for evaluating large language model (LLM) agents on the task of compiling real-world open-source C/C++ software. The authors developed BUILD-BENCH, a dataset of 148 diverse repositories, and proposed a multi-agent system called OSS-BUILD-AGENT that automatically retrieves build instructions and iteratively fixes compilation errors. Experiments show that this method outperforms both traditional rule-based and previous agentic approaches, achieving up to 71.8% success under flexible validation, demonstrating the strong potential of LLMs in automating complex software compilation tasks

**Strengths:**

1. The BUILD-BENCH benchmark covers more realistic and complex open-source projects, enabling a more comprehensive evaluation of automated compilation capabilities.
2. The proposed OSS-BUILD-AGENT leverages multi-agent collaboration and LLM-based retrieval to significantly improve compilation success rates and automatic error-repair capabilities.

**Weaknesses:**

**Unspecified Computational Costs:** The paper lacks quantitative analysis of *time cost* (the total time required to complete the entire compilation process) and *economic cost* (the total cost incurred to complete the compilation tasks), as well as a horizontal comparison with baselines—particularly **CompileAgent**.

**Insufficient Baseline Comparison:** It is recommended to include single-round data for both closed-source and open-source models: closed-source models such as Gemini-2.5-Flash, GPT-4o, and Claude-3.5-Sonnet, and open-source models such as Qwen3 235B, Qwen3 Coder 485B. Additionally,  OSS-BUILD-AGENT w/o Retrieval should also include Claude 3.7-Sonnet, Gemini-2.5-Flash, Qwen3 235B, Qwen3 Coder 485B, and a RAG-based baseline.

**Incomplete Task Information:** The paper lacks detailed information on the task set and its difficulty distribution, including details such as *project topic* and whether build instructions are **InRepo** (the repository contains a build guide), **NotInRepo** (the repository does not directly contain a build guide but external documentation is available), or **NoGuide** (the project completely lacks any build guide). The proportion of projects in each of these categories should be reported.

**Unclear Methodology:** The implementation details and prompt design of the **LLM-Assisted Retrieval Module** are not clearly described—particularly how the system retrieves other files in the repository and accesses external web resources.

**Incomplete Evaluation Metrics:** In Section 6.2, *Retrieval and Error Resolution*, the analysis of the retrieval module is overly simplistic. Evaluating retrieval success solely based on whether the retrieval module accessed the ground-truth URL that hosts the build instruction for the given repository is insufficient. The evaluation should also account for retrieval performance over other files in the repository.

**Questions:**

1.Could you explain in detail the implementation of LLM-Assisted Retrieval?

2.For lines 395–399, which describe the workflow that “*mimics a human engineer*,” could you provide a **case study** illustrating how this process is implemented in practice?

---

> ### Author Response · Authors · 2025-11-23
> **Response to Reviewer y24p (1/2)**
>
> We appreciate you finding our work as a more realistic and challenging benchmark for agents, and our agentic workflow that improves the compilation success rate significantly. Below, we address the raised concerns and specify the changes in the revised manuscript.
>
> ---
> > __Weakness 1__: Unspecified Computational Costs: The paper lacks quantitative analysis of time cost (the total time required to complete the entire compilation process) and economic cost (the total cost incurred to complete the compilation tasks), as well as a horizontal comparison with baselines—particularly CompileAgent.
>
> __A1__: To evaluate the cost of our agentic compilation method vs other baselines, we report the additional resource metrics below:
>
> We use the cost metrics that the CompileAgent paper uses. Adding together the cost of retrieval and compilation stages of OSS-Build-Agent, the average time and monetary cost is 5.27 minutes and $0.34 per repository.
>
>
> | System / Agent   | Time Cost Per Repo (Min) | LLM API Cost Per Repo (USD) |
> |-----------------------------|--------------------------|------------------------------|
> | OSS-BUILD-AGENT - GPT-4o    | 5.27                     | $0.34   |
> | CompileAgent - GPT-4o        | 6.7                      | $0.16            |
> | GHCC                         | 0.8                      | $0 (Rule-Based)              |
> | Assemblage                   | N/A                      | $0 (Rule-Based)              |
>
> A full breakdown of the cost can be found in Appendix E.
>
> ---
>
> > __Weakness 2__: Insufficient Baseline Comparison: It is recommended to include single-round data for both closed-source and open-source models: closed-source models such as Gemini-2.5-Flash, GPT-4o, and Claude-3.5-Sonnet, and open-source models such as Qwen3 235B, Qwen3 Coder 485B. Additionally, OSS-BUILD-AGENT w/o Retrieval should also include Claude 3.7-Sonnet, Gemini-2.5-Flash, Qwen3 235B, Qwen3 Coder 485B, and a RAG-based baseline.
>
> __A2__: We previously selected o3-mini and Claude-3.7-Sonnet for the “Single-Turn” and “w/o Retrieval” baseline for their competitive performance among all models.
>
> As recommended,  we tested four more models (Gemini 2.5 Flash, GPT-4o, Qwen3 235B, and Qwen3 Coder 485B) for both OSS-BUILD-AGENT “Single-Turn” and “w/o Retrieval” settings, and updated Table 1.
>
> We also added RAG as a retrieval baseline to investigate retrieval strategies’ impact on the compilation task. We report the results and performance comparison in Table 3 in Section 6.2. Detailed analysis and implementation of RAG can be found in lines 419-431.
>
>
> ---
>
> > __Weakness 3__: Incomplete Task Information: The paper lacks detailed information on the task set and its difficulty distribution, including details such as project topic and whether build instructions are InRepo (the repository contains a build guide), NotInRepo (the repository does not directly contain a build guide but external documentation is available), or NoGuide (the project completely lacks any build guide). The proportion of projects in each of these categories should be reported.
>
>
> __A3__: 136 (91.9%) of the repositories of BuildBench have build instructions available. 39 (28.7%) of such repositories have build instructions outside of the repository. We updated the information about repositories in the final compilable test set in Appendix J.
>
> > __Weakness 4__: Unclear Methodology: The implementation details and prompt design of the LLM-Assisted Retrieval Module are not clearly described—particularly how the system retrieves other files in the repository and accesses external web resources.
>
> __A4__: We would like to draw your attention to Section 3.1 and Figure 3, where we describe our implementation of LLM-Assisted Retrieval, and we added the prompt design in Appendix Section C.1. We added the following details in Section 3.1 about the tools that we implemented to improve clarity.
>
> The LLM-retrieval module can extract useful links (_both internal repository files and external web resources_) and access them respectively through tool calling (read_files and read_webpage), enabling it to navigate across documentation and online instructions. We observed the average build-information trajectory length is 2.56 in the validation test, and thus set a three iteration limit for retrieval. This upper-bound balances effective retrieval with controlled computational cost.

---

> ### Author Response · Authors · 2025-11-23
> **Response to Reviewer y24p (2/2)**
>
> > __Weakness 5__: Incomplete Evaluation Metrics: In Section 6.2, Retrieval and Error Resolution, the analysis of the retrieval module is overly simplistic. Evaluating retrieval success solely based on whether the retrieval module accessed the ground-truth URL that hosts the build instruction for the given repository is insufficient. The evaluation should also account for retrieval performance over other files in the repository.
>
> __A5__: Our retrieval method may access files inside a repository. This is because the term _URL_ that we used in the manuscript refers to both internal files and external webpages, as we would parse the internal hyperlinks embedded in documentation files to be a valid URL starts repository URL for consistency with external webpage URLs. So our retrieval method could access other files inside the repository. We clarified the definition of URL in the paper at line 395-398.
>
> To further investigate the retrieval accuracy, we measure the retrieval trajectory coverage, measuring how many intermediate URLs, which can potentially lead the retriever to the ground truth build instructions, have been captured by our retrieval modules. The results show that our retrieval module can capture around 80% of intermediate URLs or internal files, as well as the potentially useful build instructions contained in them.
>
> | Model Name           | Build Instruction URL Prediction Accuracy | Target Trajectory Coverage |
> |-----------------------|------------------------------------------|----------------------------|
> | GPT-4o               | 73.8%                                    | 81.2%                      |
> | o3-mini              | 72.8%                                    | 81%                        |
> | Gemini 2.5 Flash     | 74.3%                                    | 82.7%                      |
> | Claude 3.7 Sonnet    | 75%                                      | 84%                        |
> | Qwen3 235B           | 74.3%                                    | 82.5%                      |
> | Qwen3 Coder 485B     | 72.8%                                    | 81.2%                      |
>
> ---
>
>
> >__Question 1__: Could you explain in detail the implementation of LLM-Assisted Retrieval?
>
> __A6__: Please refer to the response to Weakness 4.
>
> ---
> >__Question 2__: For lines 395–399, which describe the workflow that “mimics a human engineer,” could you provide a case study illustrating how this process is implemented in practice?
>
> __A7__:We clarify here by “mimic human engineer” we mean that our retrieval modules are explicitly designed not to access build scripts like Makefile or CMakeLists.txt, which are often noisy. Retrieving prepared build instructions is an efficient and intuitive way for human engineers to complete the compilation task, and this is also the main motivation of designing our retrieval module.

---

> ### Author Response · Authors · 2025-11-28
>
> I hope this message finds you well. As the discussion period is ending in less than a week. I wanted to ensure we have addressed all your concerns fully. If there are any points or feedback you'd like us to address, please let us know. Your input and feedback are valuable to us to improve our work. We appreciate your time and effort in reviewing our paper.

---

### Author Response · Authors · 2025-12-04
**Rebuttal Summary**

Dear Reviewers and ACs


Due to the unfortunate interruption of the rebuttal process, we were unable to receive further feedback from the reviewers. To document the discussion, we summarize and categorize the reviewers’ concerns and our responses below for future reference. We sincerely appreciate the reviewers’ time and effort in providing thoughtful feedback that has helped improve our work.

---

>**1. Benchmark realism, construction, and scope**

**Concerns**: Are the build failures realistic? Do human experts introduce data selection bias? Why only Linux and C/C++?

**Responses**: We clarify that all repositories are cloned and kept intact before conducting compilation methods, with no synthetic errors injected. We describe the criteria that human labelers used to determine whether a repository is “uncompilable,” and we release these “uncompilable” repositories as an optional hard-mode dataset. We motivate our focus on Linux C/C++ repositories through downstream binary-analysis use cases and add discussion of other OS/language support to the Limitations section.

---
>**2. Retrieval design, baselines, and ablations**

**Concerns**: Limited baselines; limited ablation study of the LLM-Assisted Retrieval module (base model sensitivity, comparison with heuristic methods, RAG); insufficient elaboration on the retrieval module details.

**Responses**: We add more LLM baseline experiments (including single-turn and no-retrieval settings), as well as TF-IDF (a lightweight heuristic) and RAG retrieval baselines for the retrieval task. We also add ablation experiments comparing No Retrieval, RAG, our LLM retriever, and a Perfect Retrieval as upper bound on the end-to-end compilation task, demonstrating that retrieval quality is a key bottleneck and that our method offers clear advantages. We further clarify and elaborate the retrieval workflow, tools, link-following behavior, and depth choices.

---
>**3. Evaluation metrics and analysis**

**Concerns**: Evaluation metrics are binary (success or failure); retrieval and failure mode analysis should be further elaborated.

**Responses**: We reiterate the necessity of using validated success (strict/flexible) to eliminate false positives and have added pass@K and number-of-attempts metrics to capture gradual repair. For retrieval, we clarify that URLs include both external documentation and in-repository files (GitHub URLs) and introduce retrieval-trajectory coverage analysis. We further extend the failure-mode analysis with additional breakdowns and case studies.

---
>**4. Efficiency and practicality**

**Concerns**: Lack of runtime, token, and cost analysis versus prior agents and rule-based systems.

**Responses**: We compare time and computational cost (tokens) for our agent, the agentic baseline, and rule-based baselines, and we provide a more detailed breakdown (including retrieval costs) in the appendix, clarifying the performance–cost trade-offs.

---

### Meta-Review · Area_Chair_c7aL · 2026-01-07

**Summary:**

This submission receives original ratings of 6, 4, 4, 4. The reviewer consensus is that this work presents a well engineered benchmark. However, they also raises concerns regarding the design and demonstration of this bench, that is, its evaluation analysis and metric design, which  is critical to tell the audience about the development / intelligence of the LLMs. The reviewers also raise other issues but they are not the tie breaker for the decision recommendation. Essentially, the LLM benchmark works are really important and fashioned, and there are many of them both in the ICLR submissions and in the community. It is then comes down to whether this proposed benchmark tell us something about the frontier models and which part of the LLMs' intelligence can be further improved.

**Reviewer Concerns:**

Concerns that are mostly addressed:
1. missing computational cost;
2. added RAG and single turn baselines;
3. task set stats and breakdown;
4. clarification on details and limitations;

Concerns that are not fully addressed:
1. retrieval evaluation;
2. coarse evaluation process and metrics;
3. efficiency metrics;

**Reviewer Scores:**

Reviewer y24p (4) would likely raise its rating as most of detailed comments are addressed and there are no major concerns left.

Reviewer tmyd (4) most likely to keep rating as the metric concern softens but not disappears.

Reviewer z2nx (6) most likely to keep rating as the efficiency metric concern softens but not disappears.

Reviewer atpj (4) most likely to keep its rating. The concern is on evaluation choice, and marginal improvements and stronger baselines. Thought the rebuttal reframes the contribution of this submission but it is difficult to address this type of questions (as always).

---

### Decision · Program_Chairs · 2026-01-26

Reject